# Efficient Leverage Score Sampling for Tensor Train Decomposition

**Vivek Bharadwaj**[*,1] **, Beheshteh T. Rakhshan**[*,2] **, Osman Asif Malik**[3] **, Guillaume Rabusseau**[2,4]

[1]Electrical Engineering and Computer Science Department, UC Berkeley
[3]Encube Technologies
[2]Mila & DIRO, Université de Montréal
[4]CIFAR AI Chair

## Abstract

Tensor Train (TT) decomposition is widely used in the machine learning and quantum physics communities as a popular tool to efficiently compress high-dimensional tensor data. In this paper, we propose an efficient algorithm to accelerate computing the TT decomposition with the Alternating Least Squares (ALS) algorithm relying on exact leverage scores sampling. For this purpose, we propose a data structure that allows us to efficiently sample from the tensor with time complexity logarithmic in the tensor size. Our contribution specifically leverages the canonical form of the TT decomposition. By maintaining the canonical form through each iteration of ALS, we can efficiently compute (and sample from) the leverage scores, thus achieving significant speed-up in solving each sketched least-square problem. Experiments on synthetic and real data on dense and sparse tensors demonstrate that our method outperforms SVD-based and ALS-based algorithms.

## 1 Introduction

Tensor decomposition methods have recently found numerous applications in machine learning. Their ability to perform operations efficiently on very high-dimensional tensors makes them suitable for data science and machine learning problems. For example, they have been used for neuro-imaging, and signal processing [Zhou et al., 2013, Sidiropoulos et al., 2017, Cichocki and Phan, 2009], supervised learning [Novikov et al., 2016, Stoudenmire and Schwab, 2016], feature extraction [Bengua et al., 2015] and scaling up Gaussian processes [Izmailov et al., 2018]. The most popular decompositions are the CANDECOMP/PARAFAC (CP), Tucker [Hitchcock, 1927, Tucker, 1966], Tensor Train (TT) Oseledets [2011] and Tensor Ring (TR) Zhao et al. [2016] decompositions. However, finding a rank-$r$ CP decomposition is NP-hard [Kolda and Bader, 2009, Hillar and Lim, 2013] and in the Tucker decomposition, the number of parameters grows exponentially with the order of the tensor. While the number of parameters is linear in the tensor's order for both TT and TR decompositions, TR is known to have numerical stability issues. The TT decomposition, in contrast, can be used to represent a tensor in a compressed format where the number of parameters scales linearly with the order of a tensor. Additionally, finding a good approximation to the best rank-$r$ TT decomposition is feasible which lets the user perform various linear algebra operations efficiently and compute the TT decomposition with stable algorithms.

Due to the high-dimensional nature of tensors, designing efficient algorithms for computing the TT decomposition is crucial. A popular method for computing the TT decomposition of an $N$-dimensional tensor $\mathcal{X}$ is the TT-SVD algorithm [Oseledets, 2011] which uses a sequence of singular values decompositions on the tensor unfoldings to produce the TT representation in a single pass.

---

[*]Equal contribution

38th Conference on Neural Information Processing Systems (NeurIPS 2024).

Since TT-SVD requires performing SVDs of unfoldings of $\mathcal{X}$, its cost is exponential in $N$. Alternating Least Square (ALS) is another popular approach [Holtz et al., 2012] to find the TT approximation. Starting with a crude guess, each iteration of ALS involves solving a sequence of least squares problems. While ALS is the workhorse algorithm in many tensor decomposition problems, the computational cost is still exponential in the order of a tensor ($N$), since each iteration requires solving least squares problems involving unfoldings of $\mathcal{X}$. These issues have led to the search for alternatives based on randomization and sampling techniques. A cheaper alternative to the TT-SVD with strong accuracy guarantees can be implemented by replacing the exact singular value decomposition (SVD) with a well-studied randomized counterpart [Halko et al., 2011, Huber et al., 2017]. Randomized variants of the TT-ALS approach have received little attention. In Chen et al. [2023], the authors propose a randomized ALS algorithm that uses TensorSketch [Pham and Pagh, 2013] in each iteration. In this work, we also propose a novel randomized variant of the TT-ALS algorithm that relies on exact leverage score sampling. Notably, the sketch size in TensorSketch TT-ALS Chen et al. [2023] has an exponential dependence on the tensor dimension $I$ whereas our algorithm avoids any dependence of the sketch size on $I$.

**Our Contributions.** In this paper, we propose a new sampling-based ALS approach to compute the TT decomposition: rTT-ALS. By using exact leverage score sampling, we are able to significantly reduce the size of each ALS least squares problem while providing strong guarantees on the approximation error. At the core of rTT-ALS, we leverage the TT canonical form to efficiently compute the exact leverage scores and speed up the solutions of least square problems in each iteration of ALS. To the best of our knowledge, rTT-ALS is the first efficient TT decomposition by the ALS algorithm which relies on leverage scores sampling. We provide experiments on synthetic and real massive sparse and dense tensors showing that rTT-ALS can achieve up to $26\times$ speed-up compared to its non-randomized counterpart with little to no loss in accuracy.

Our core contribution is the following theorem, which shows that we can efficiently compute a subspace embedding of a left-orthogonal chain of TT tensor cores by efficiently sampling according to their squared row norms (for technical definitions and details see subsection 3.1).

**Theorem 1.1** (Row-norm-squared sampling for 3D core chains). *Let $\mathcal{A}_1, ..., \mathcal{A}_j$ be a sequence of 3D tensors, $\mathcal{A}_k \in \mathbb{R}^{R_{k-1} \times I_k \times R_k}$ (with $R_0 = 1$). Assume that the left-matricization of each core is orthogonal. Let $A_{\leq j}$ be the $\prod_{k=1}^{j} I_k \times R_j$ matrix obtained by unfolding the contraction of the tensor chain $\mathcal{A}_1, ..., \mathcal{A}_j$. Then there exists a data structure to randomly sample rows from $A_{\leq j}$ according to the distribution of its squared row norms with the following properties:*

1. *The data structure has construction time $O\left(\sum_{n=1}^{j} I_n R_{n-1} R_n^2\right)$. When $R = R_1 = ... = R_j$ and $I = I_1 = ... = I_j$, the runtime is $O(jIR^3)$. The space overhead of the data structure is linear in the sizes of the input cores.*

2. *The data structure produces a single row sample from $A_{\leq j}$ according to the distribution of its squared row norms in time $O\left(\sum_{k=1}^{j} \log\left(I_k R_{k-1}/R_k\right) R_k^2\right)$. When all ranks $R_k$ and physical dimensions $I_k$ are equal, this complexity is $O(jR^2 \log I)$.*

We highlight that the runtime required to construct the data structure is asymptotically identical to the runtime required to compute the canonical form of the tensor train subchain, i.e., $A_{\leq j}$, by successive QR decompositions. This implies that the data structure construction and subsequent updates of the data structure do not increase the asymptotic complexity of our method.

## 2 Related work

Randomized algorithms and leverage score sampling-based methods [Mahoney et al., 2011, Woodruff et al., 2014, Drineas et al., 2006a] have been used widely in a large body of research particularly in tensor decomposition problems over the past two decades [Malik and Becker, 2021, Larsen and Kolda, 2022, Fahrbach et al., 2022] just to name a few.

[Cheng et al., 2016] propose SPALS, the first ALS-based algorithm relying on leverage score sampling for the CP decomposition. Their proposed method reduces the size of the least squares problem in each iteration of ALS with a sub-linear cost per iteration in the number of entries of the input

tensor. Larsen and Kolda [2022] extends this method by combining repeated sampled rows in a deterministic and random sampling fashion. However, both of these methods use leverage score approximations and therefore require a number of samples which is exponential in the number of tensor modes in order to achieve relative-error performance guarantees. Malik [2022] proposes a method which avoids this exponential dependency on the number of tensor modes by using higher-quality leverage score estimates for the CP decomposition. The method is further improved by [Malik et al., 2022] to use exact rather than approximate leverage scores which is applicable for arbitrary tensor decompositions. Recently, [Bharadwaj et al., 2023] provided a novel data structure to efficiently sample from the exact distribution of the factor matrices' leverage scores in the Khatri-Rao product with time complexity logarithmic in the tensor size, leading to further improvements on the work in [Malik et al., 2022]. Moreover, [Malik and Becker, 2021] proposed an ALS-based algorithm for finding the TR decomposition using leverage scores approximation. However, the runtime of their method has an exponential dependency on the order of a tensor. The sampler we propose in this paper is built on the work by Bharadwaj et al. [2023], extending it to the TT decomposition and leveraging the canonical form for further speed-up.

There are also a variety of non-ALS-based randomized algorithms for computing the TT decomposition. [Huber et al., 2017] leverages randomized SVD for the TT decomposition which accelerates the classical TT-SVD algorithm proposed by [Oseledets, 2011]. To handle situations where the exact TT rank is unknown, [Che and Wei, 2019] propose an adaptive randomized algorithm that can achieve near optimal TT approximation. [Yu et al., 2023] present a method leveraging randomized block Krylov subspace iteration for computing TT approximations. Most of the algorithms for TT decomposition are based on the randomized SVD for matrices introduced by [Halko et al., 2011]. In the quantum physics community, the ALS algorithm is widely used for finding TT decomposition and often yields more accurate results than TT-SVD. The randomized TT-SVD method struggles to scale for high-order tensors as it requires generating a random Gaussian matrix at each step that can only handle small-order tensors. More closely related to our work are those using sketching and sampling in each iteration of ALS to approximate the TT decomposition. Recently, [Chen et al., 2023] introduced an algorithm that employs TensorSketch [Pham and Pagh, 2013] in each iteration of a regularized ALS approach for TT decomposition. However, the sketch size has an exponential dependency to a tensor dimension. By contrast, our proposed algorithm requires the sketch size with no dependence on the tensor dimension and it depends only on the column size of the design matrix and $\varepsilon$ and $\delta$ parameters.

## 3  Preliminaries

We use capital letters $A$ to denote matrices and script characters $\mathcal{A}$ to denote multidimensional arrays. We use Matlab notation for slices of matrices and tensors. We use the tuple notation to indicate the position of entries of arrays. For example, $\mathcal{A}(i_1, i_2, i_3)$ indicates the $(i_1, i_2, i_3)$-th element of $\mathcal{A}$. $A[i, :]$ and $A[:, i]$ refer to the $i$-th row and column of $A$, respectively; for a three-dimensional tensor $\mathcal{A} \in \mathbb{R}^{R_1 \times I_1 \times R_2}$, the matrix $\mathcal{A}[:, i, :] \in \mathbb{R}^{R_1 \times R_2}$ is the $i$-th lateral slice of $\mathcal{A}$. For a positive integer $n$, we use $[n]$ to denote the set of integers from 1 to $n$. For $i_1 \in [I_1], \ldots, i_N \in [I_N]$, the notation $\underline{i_1 \ldots i_N} \overset{\text{def}}{=} 1 + \sum_{n=1}^{N} (i_n - 1) \prod_{j=1}^{N-1} I_j$ will be helpful for tensor unfoldings. We use $\otimes$ and $\odot$ to denote the Kronecker and Khatri-Rao products, respectively (see definitions in Appendix A). We use $I_d$ to denote the $d \times d$ identity matrix, $A^{\mathsf{T}}$ for the transpose of $A$, $A^+$ for the pseudo-inverse of $A$, $\|\cdot\|_F$ for the Frobenius norm and $\|\cdot\|_2$ for the Euclidean norm of a vector. We use $\tilde{O}$ to indicate the presence of multiplicative terms polylogarithmic in $R$ and $1/\delta$.

### 3.1  Tensor Train Decomposition

Let $\mathcal{X} \in \mathbb{R}^{I_1 \times \cdots \times I_N}$ be an $N$-dimensional array. A rank $(R_1, \ldots, R_{N-1})$ *tensor train (TT) decomposition* of a tensor $\mathcal{X} \in \mathbb{R}^{I_1 \times \cdots \times I_N}$ factorizes it into the product of $N$ third-order tensors $\mathcal{A}_n \in \mathbb{R}^{R_{n-1} \times I_n \times R_n}$ for $n \in [N]$ (with $R_0 = R_N = 1$):

$$\mathcal{X}(i_1, \cdots, i_N) = \sum_{r_0, \cdots, r_N} \prod_{n=1}^{N} \mathcal{A}_n(r_{n-1}, i_n, r_n),$$

for all $i_1 \in [I_1], \cdots, i_N \in [I_N]$, where each $r_n$ ranges from 1 to $R_n$. A tensor network representation of a TT decomposition is shown in Figure 1. We call $\mathcal{A}_1, \mathcal{A}_2, \cdots, \mathcal{A}_N$ core tensors and we use $\mathrm{TT}((\mathcal{A}_n)_{n=1}^N)$ to denote a TT tensor with factors $\mathcal{A}_1, \cdots, \mathcal{A}_n$.

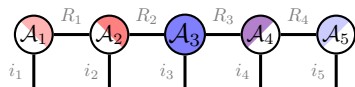

Figure 1: Tensor Train decomposition of a 5-dimensional tensor in tensor network notation.

**Definition 3.1.** *The mode-$n$ unfolding of a tensor $\mathcal{X} \in \mathbb{R}^{I_1 \times \cdots \times I_N}$ is the matrix $X_{(n)} \in \mathbb{R}^{I_n \times \prod_{j \neq n} I_j}$ defined element-wise by $X_{(n)} \left( i_n, \underline{i_1 \cdots i_{n-1} i_{n+1} \cdots i_N} \right) \stackrel{\text{def}}{=} \mathcal{X}(i_1, \cdots, i_N)$.*

*As a special case, we denote the left (resp. right) matricization of a 3-dimensional tensor $\mathcal{A} \in \mathbb{R}^{I_1 \times I_2 \times I_3}$ by $A^L = (A)_{(3)}^\top \in \mathbb{R}^{I_1 I_2 \times I_3}$ and $A^R = A_{(1)} \in \mathbb{R}^{I_1 \times I_2 I_3}$.*

Given a TT decomposition $\mathrm{TT}((\mathcal{A}_n)_{n=1}^N)$ and an index $j$, we will often use the left-chain $A_{<j} \in \mathbb{R}^{\prod_{k=1}^{j-1} I_k \times R_{j-1}}$ and right-chain $A_{>j} \in \mathbb{R}^{R_j \times \prod_{k=j+1}^N I_k}$ unfoldings obtained by matricizing the contraction of all cores on the left and on the right side of the $j$-th core. Formally,

$$A_{<j}(i_{<j}, r_{j-1}) = \sum_{r_0, \ldots, r_{j-1}} \prod_{k=1}^{j-1} \mathcal{A}_k(r_{k-1}, i_k, r_k) \text{ and } A_{>j}(r_j, i_{>j}) = \sum_{r_{j+1}, \ldots, r_N} \prod_{k=j+1}^{N} \mathcal{A}_k(r_{k-1}, i_k, r_k)$$

where $i_{<j} = i_1 \ldots i_{j-1}$ and $i_{>j} = i_{j+1} \ldots i_N$. We also use $A^{\neq j} \stackrel{\text{def}}{=} A_{<j} \otimes A_{>j}^\top \in \mathbb{R}^{\prod_{k \neq j} I_k \times R_{j-1} R_j}$ to denote the unfolding of the contraction of all cores except the $j$-th one.

We conclude by introducing the canonical form of the TT decomposition [Holtz et al., 2012, Evenbly, 2018, 2022] which will be central to the design of our algorithm.

**Definition 3.2.** *A TT decomposition $\mathrm{TT}((\mathcal{A}_n)_{n=1}^N) \in \mathbb{R}^{I_1 \times \cdots \times I_N}$ is in a canonical format with respect to a fixed index $j \in [N]$ if $A_n^{L^\top} A_n^L = I_{R_n}$ for all $n < j$, and $A_n^R A_n^{R^\top} = I_{R_{n-1}}$ for all $n > j$ (see Figure 2).*

Figure 2: Orthonormal TT decomposition. The cores at the left side of $\mathcal{A}_3$ are left-orthonormal and the cores at the right are right-orthonormal.

Note that any TT decomposition can efficiently be converted to canonical form w.r.t. any index $j \in [N]$ by performing a series of QR decompositions on the core tensors [Holtz et al., 2012, Evenbly, 2018].

## 3.2 Alternating Least Squares with Tensor Train Structure.

The TT decomposition problem consists in finding a low-rank approximation $\mathrm{TT}((\mathcal{A}_n)_{n=1}^N)$ of a given tensor $\mathcal{X}$: $\mathrm{argmin}_{\mathcal{A}_1, \ldots, \mathcal{A}_N} \|\mathcal{X} - \mathrm{TT}(\mathcal{A}_1, \ldots, \mathcal{A}_N)\|_F$ where $\mathcal{X}$ is the target tensor with dimensions $I_1 \times \cdots \times I_N$. Since this is a non-convex optimization problem, the popular alternating least-squares (ALS) approach can be used to find an approximate solution [Kolda and Bader, 2009]. Fixing all cores except the $j$-th one, the low rank approximation problem can be reformulated as a linear least squares problem:

$$\mathrm{argmin}_{\mathcal{A}_j} \left\| \left( A_{<j} \otimes A_{>j}^\top \right) (A_j)_{(2)}^\top - X_{(j)}^\top \right\|_F. \tag{1}$$

The ALS approach finds an approximate solution by keeping all cores fixed and solving for the $j$-th one. Then repeat this procedure multiple times for each $j \in [N]$ until some convergence

criteria is met. While ALS is the workhorse algorithm in many tensor decomposition problems, the computational cost is still $O(I^N)$ for $I_1 = \cdots = I_N = I$. In this work, to reduce this cost, we will combine ALS with core orthogonalization to efficiently compute the exact leverage scores. This will also lead to a stable algorithm for computing TT. To compute the orthogonalized TT approximation, we start with a crude TT decomposition in canonical form (see Definition 3.2) where all cores except the first one are right-orthonormal. After optimizing the first core, a QR decomposition is performed and the non-orthonormal part is merged into the next core. This procedure repeats until reaching the right side of the decomposition. The same procedure is then repeated from the right until reaching the left side (see the tensor network illustration in Appendix A.1). Even after computing the canonical form, which makes the design matrix orthonormal for each linear least squares problem, we still need to multiply the matricized tensor by the chain of TT cores in canonical form. Given that the tensor may have hundreds of millions of nonzeros, this multiplication is a significant computational bottleneck: without sketching, the runtime for this matrix multiplication scales as $O(\text{nnz}(\mathcal{X})NR^2)$, where $\text{nnz}(\mathcal{X})$ is the number of nonzeros in the tensor. Sketching allows us to select only a subset of rows from the design matrix and the corresponding subset of rows from the matricized tensor, reducing the cost to $\tilde{O}(NR^2)$. This approach also leads to providing computational benefits for computing the leverage scores and to an efficient sampling scheme which will be discussed in Section 4.

### 3.3 Sketching and Leverage Score Sampling

There exists a vast literature on randomized algorithms [Mahoney et al., 2011, Woodruff et al., 2014] to solve the over-determined least squares problem $\min_x \|Ax - b\|_F$ where $A \in \mathbb{R}^{I \times R}, I \gg R$. Regardless of the structure of both $A$ and $b$, solving this least-squares problem costs $O(IR^2)$. To reduce this cost, we can randomly select rows of $A$ and $b$ by proposing a sketching operator $S$ with $J \ll I$. Therefore, instead of solving the original least squares problem, we consider solving the downsampled version of the form $\min_x \|SAx - Sb\|_F$, where $S \in \mathbb{R}^{J \times I}$ and reduce the cost to $O(JR^2)$. The goal is to find a "good" sketch $S$ to approximate the solution of the least squares problem at each step of the ALS algorithm. When each entry of $S$ is selected according to the rows of $A$ leverage scores, strong guarantees can be obtained for the solution of the downsampled problem.

**Definition 3.3.** *(Leverage scores) Suppose $A \in \mathbb{R}^{I \times R}$ with $I \gg R$. The $i$-th leverage score of the matrix $A$ is defined as*

$$l_i(A) = A[i,:](A^\top A)^+ A[i,:]^\top \text{ for } i \in [I]. \tag{2}$$

**Definition 3.4.** *(Leverage score sampling) Let $A \in \mathbb{R}^{I \times R}$ and $p \in [0,1]^I$ be a probability distribution vector with entries $p_i = \frac{l_i(A)}{rank(A)}$; where $rank(A) = \sum_i l_i(A)$. Assume $\hat{s}_1, ..., \hat{s}_J$ are drawn i.i.d according to the probabilities $p_1, \cdots, p_I$. The random matrix $S \in \mathbb{R}^{J \times I}$ defined element-wise by $S(j,i) = \frac{1}{\sqrt{Jp_i}}$ if $\hat{s}_j = i$ and 0 otherwise is called a leverage score sampling matrix for $A$.*

The following result is well-known and appeared in several works; see, e.g., [Drineas et al., 2006b], [Drineas et al., 2008], [Drineas et al., 2011], [Larsen and Kolda, 2022]. We borrow the form presented in [Malik, 2022].

**Theorem 3.5.** *(Guarantees for Leverage Score Sampling) Suppose $A \in \mathbb{R}^{I \times R}$. Let $S \in \mathbb{R}^{J \times I}$ be the leverage score sampling matrix defined in 3.4. For any $\varepsilon, \delta \in (0,1)$, if $J = \tilde{O}(R^2/\varepsilon\delta)$, then $\tilde{x}^* = \min_x \|SAx - Sb\|_2$ satisfies $\|Ax^* - b\|_2 \le (1+\varepsilon)\min_x \|Ax - b\|_2$, with probability $1 - \delta$.*

Computing leverage scores in Definition 3.3 requires computing the pseudo-inverse of $A$, which costs $O(IR^2)$ and is as costly as directly solving the original least squares problem. In the following section, we will show that the leverage scores can be computed much more efficiently when $A$ is the matrix appearing in the TT-ALS algorithm in canonical form..

## 4 Sampling-based Tensor Train Decomposition

In this section, we show how to efficiently sample rows of $A^{\ne j} = A_{<j} \otimes A_{>j}^\top$ and $X_{(j)}$ in Equation (1) according to the exact leverage scores distribution. In doing so, we will also present the sketch of the proof of Theorem 1.1 (which closely mirrors that of Bharadwaj et al. [2023] with key modifications required to adapt the procedure to a tensor core chain).

For each row $i^{\neq j} = \underline{i_1 \ldots i_{j-1} i_{j+1} \ldots i_N}$ of $A^{\neq j}$, Equation (2) gives

$$l_{i^{\neq j}}(A^{\neq j}) = A^{\neq j}[i^{\neq j}, :](A^{\neq j^\top} A^{\neq j})^+ A^{\neq j}[i^{\neq j}, :]^\top. \tag{3}$$

Computing $\Phi \overset{\text{def}}{=} (A^{\neq j^\top} A^{\neq j})^+$ is the main computational bottleneck in finding the leverage scores of $A^{\neq j}$. Malik et al. [2022] proposed an algorithm to compute $\Phi$ in time $O(NIR^2 + R^3)$. In this paper, we leverage the fact that when the TT tensor is in canonical form w.r.t. mode $j$, $A^{\neq j}$ is orthogonal, and thus $\Phi = I_{R^2}$. Therefore, computing $\Phi$ is free of cost. By maintaining the canonical form of the TT tensor throughout the ALS algorithm, we can sketch the least square problems from the leverage score distributions with almost no computational overhead. We now explain how to efficiently sample rows of $A^{\neq j}$ from the leverage scores distribution.

## 4.1 Efficient Core Chain Leverage Score Sampling

As discussed above, when the TT tensor is in canonical form, the leverage score of row $i^{\neq j}$ is given by $l_{i^{\neq j}}(A^{\neq j}) = A^{\neq j}[i^{\neq j}, :]A^{\neq j}[:, i^{\neq j}]^\top$. Leveraging the Kronecker structure of $A^{\neq j} = A_{<j} \otimes A_{>j}^\top$, one can easily show that $l_{i^{\neq j}}(A^{\neq j}) = l_{i_{<j}}(A_{<j}) \cdot l_{i_{>j}}(A_{>j}^\top)$. Sampling from the leverage scores distributions thus boils down to sampling rows of $A_{<j}$ and $A_{>j}^\top$ with probability proportional to their squared row norms (due to the orthogonality of $A_{<j}$ and $A_{>j}$ inherited from the canonical form). Without loss of generality, we detail the sampling procedure for $A_{\leq j}$ (the difference between $A_{\leq j}$ and $A_{<j}$ amounts to reindexing). The sampling procedure for $A_{>j}$ will be the same and straightforward.

Let $\hat{s}_1 \in [I_1], ..., \hat{s}_j \in [I_j]$ be random variables such that the multi-index $\hat{s}_{\leq j} = \underline{\hat{s}_1 \ldots \hat{s}_j}$ follows the leverage score distribution of $A_{\leq j}$. Since $\mathrm{TT}((\mathcal{A}_n)_{n=1}^N)$ is in canonical form w.r.t. $j + 1$, $A_{\leq j}$ is an orthonormal matrix, hence $\underline{\hat{s}_1 \ldots \hat{s}_j}$ is selected with probability proportional to the squared norm of the corresponding row of $A_{\leq j}$:

$$p(\hat{s}_1 = s_1, \ldots, \hat{s}_j = s_j) := \frac{1}{R_j} \left( A_{\leq j}[\underline{s_1 \ldots s_j}, :] \cdot A_{\leq j}[\underline{s_1 \ldots s_j}, :]^\top \right). \tag{4}$$

Our sampling procedure will draw a lateral slice from each core starting from $\mathcal{A}_j$ and ending with $\mathcal{A}_1$, corresponding to a single row of $A_{\leq j}$. Suppose we have drawn $s_{k+1}, \ldots, s_j$, for some $k < j$. To sample the $k$-th index, we need to compute the conditional probability $p(s_k | s_{k+1}, \ldots, s_j) = \frac{p(s_k, \ldots, s_j)}{p(s_{k+1}, \ldots, s_j)}$. The following lemma shows that this can be done efficiently by leveraging the underlying TT structure.

**Lemma 4.1** (Conditional distribution for $\hat{s}_k$). *Consider the events $\hat{s}_j = s_j, \ldots, \hat{s}_{k+1} = s_{k+1}$, which we abbreviate as $\hat{s}_{>k} = s_{>k}$. Then*

$$p(\hat{s}_k = s_k \mid \hat{s}_{>k} = s_{>k}) \propto \mathrm{Tr}\left[ H_{>k}^\top \cdot \mathcal{A}_k[:, s_k, :]^\top \cdot \mathcal{A}_k[:, s_k, :] \cdot H_{>k} \right],$$

*where $H_{>k} := \mathcal{A}_{k+1}[:, s_{k+1}, :] \cdot \ldots \cdot \mathcal{A}_j[:, s_j, :]$.*

The proof is given in Appendix B. Intuitively, $H_{>k}$ acts as a "history matrix" conditioning on $s_{>k}$, while the trace operation corresponds to marginalization over $s_{<k}$. Unfortunately, updating $H_{>k}$ through matrix multiplication as each index is selected still requires time $O(R^3)$ (assuming $R_1 = \ldots = R_j = R$). In order to further improve the runtime and reach the quadratic complexity in $R$ claimed in Theorem 1.1, we make the following observation: let $q \in \mathbb{R}^{\prod_{i \leq j} I_i}$ be the probability vector for the leverage score distribution of $A_{\leq j}$. Then Equation (4) can be rewritten in vector form as $q := \frac{1}{R_j} \left( A_{\leq j}[:, 1]^2 + \ldots + A_{\leq j}[:, R_j]^2 \right)$. Here, the square of each column vector is an elementwise operation. Observe that each $A_{\leq j}[:, r]^2$ is a probability vector (positive entries summing to one) due to the orthonormality of $A_{\leq j}$. Hence $q$ is a *mixture distribution*. To sample from $q$, it thus suffices to select a single column $\hat{r}$ of $A_{\leq j}$ uniformly at random and restrict the sampling procedure to $A_{\leq j}[:, \hat{r}]^2$. More formally, let $\hat{r}$ be uniformly distributed over $[R_j]$ and let $\hat{t}_1, ..., \hat{t}_j$ follow the conditional distributions defined by

$$p(\hat{t}_k = t_k \mid \hat{t}_{k+1} = t_{k+1}, \ldots, \hat{t}_j = t_j, \hat{r} = r) = \|\mathcal{A}_k[:, t_k, :] \cdot h_{>k}\|^2, \tag{5}$$

where $h_{>k} = \mathcal{A}_{k+1}[:, t_{k+1}, :] \cdot \ldots \cdot \mathcal{A}_j[:, t_j, r]$. We have the following result.

**Lemma 4.2.** *For any choice of $s_j, ..., s_k$, fix $s_j = t_j, s_{j-1} = t_{j-1}, ..., s_k = t_k$. After marginalizing over $\hat{r}$, the conditional distribution of $\hat{t}_k$ satisfies $p(\hat{t}_k = t_k \mid \hat{t}_{>k} = t_{>k}) = p(\hat{s}_k = s_k \mid \hat{s}_{>k} = s_{>k})$.*

As a consequence, the joint random variable $(\hat{t}_1, ..., \hat{t}_j)$ follows the desired squared row-norm distribution of $A_{\leq j}$ after marginalizing over $\hat{r}$. The proof appears in Appendix B.2. Notice that the "history matrix" $H_{>k}$ has been replaced by a vector $h_{>k}$. This vector can be updated by matrix-vector multiplication, yielding a reduced sampling complexity with only a quadratic dependency on $R$.

Our final improvement is to show that each sample from the distribution in Equation (5) can be drawn in time sublinear in the dimension $I_k$ (after appropriate preprocessing). Letting $A_k^L$ be the left unfolding of $\mathcal{A}_k$, one can check that

$$p(\hat{t}_k = t_k \mid \hat{t}_{>k} = t_{>k}, \hat{r} = r) = \sum_{i=0}^{R_{k-1}-1} \left( A_k^L \left[ t_k R_{k-1} + i, : \right] \cdot h_{>k} \right)^2. \tag{6}$$

The probability of selecting the slice $s_k$ is thus the sum of $R_{k-1}$ consecutive entries from the probability vector $(A_k^L \cdot h_{>k})^2$. As a result, we can sample $\hat{t}_k$ by first sampling an index in the range $[I_k R_{k-1}]$ given by $(A_k^L \cdot h_{>k})^2$, then performing integer division by $R_{k-1}$ to obtain the corresponding slice index $\hat{t}_k$. The advantage here lies in an efficient data structure for sampling from the weight vector $(A_k^L \cdot h_{>k})^2$, given by the following lemma:

**Lemma 4.3** (Bharadwaj et al. [2023], Adapted)**.** *Given a matrix $A \in \mathbb{R}^{I \times R}$, there exists a data structure with construction time $O(IR^2)$ and space usage $O(IR)$ such that, given any vector $h \in \mathbb{R}^R$, a single sample from the un-normalized distribution of weights $(A \cdot h)^2$ can be drawn in time $O(R^2 \log(I/R))$.*

The adaptation of this lemma is given in Appendix B.3. Lemma 4.3 enables us to efficiently draw samples according to the distribution in Equation 6, and therefore gives us a procedure to sample from the entire core chain. Constructing the data structure above for each matrix $A_k^L$, $1 \leq k \leq j$, costs $O(IR_{k-1}R_k^2)$ with a linear space overhead in the input core sizes. Drawing a sample from the $k$-th data structure requires time $O(R_k^2 \log(I_k R_{k-1}/R_k))$. Summing up this runtime over $1 \leq k \leq j$ gives the stated complexity in Theorem 1.1. Algorithms 1 and 2 summarize the procedures to efficiently draw $J$ samples from a left-orthogonal core chain. The construction procedure builds a set of data structures $Z_k$ of the form given by Lemma 4.3 on the left-matricization of each tensor core. For each

---

**Algorithm 1** ConstructChainSampler($\mathcal{A}_1, ..., \mathcal{A}_N$)

1: **for** $k = 1..N$ **do**
2:     $Z_k := \text{BuildSampler}(A_k^L)$

---

**Algorithm 2** ChainSampleLeft($J, j$)

1: **for** $d = 1..J$ **do**
2:     $\hat{r} := \text{Uniform-sample}([1...R_j])$
3:     $h := e_{\hat{r}}$
4:     **for** $k = j...1$ **do**
5:         $\hat{t}_k := \text{RowSample}(Z_k, h)//R_{k-1}$
6:         $h = h \cdot \mathcal{A}_k \left[ :, \hat{t}_k, : \right]$
7:     $t_d = (\hat{t}_k)_{k \leq j}$
8: **return** $t_1, ..., t_J$

---

of $J$ rows to draw, the sampling algorithm selects a column $\hat{t}$ uniformly at random from the left matricization. It then initializes the history vector $h$ and successively samples indices $\hat{t}_{j-1}, ..., \hat{t}_1$ according to the conditional distribution, updating the history vector at each step. Appendix B.4 provides a rigorous proof of the correctness of the procedure sketched in this section.

While our procedure shares similarities with the Khatri-Rao product leverage score sampler, significant adaptations are required to sample from a tensor train core chain. The factors of a Khatri-Rao product can be sampled in any order, since the Khatri-Rao product of several matrices is commutative up to a permutation of its rows. By contrast, our sampling procedure **requires** us to sample from core $\mathcal{A}_j$ down to $\mathcal{A}_1$, since Lemma 4.1 exploits the left-orthogonality of the each core in its derivation. Starting the sampling procedure at $\mathcal{A}_j$ leads to a "history matrix" to keep track of prior draws instead of the vector that would arise starting from core $\mathcal{A}_1$. Here, our second innovation of sampling a column uniformly at random is required to bring down the overall sampling complexity. We can now state the following guarantee for **Randomized-TT-ALS** (rTT-ALS) applying the data structure in Theorem 1.1. The proof is given in Appendix B.5.

**Corollary 4.4.** *(rTT-ALS) For any $\varepsilon, \delta \in (0,1)$ the sampling procedure proposed above guarantees that with $J = \tilde{O}(R^2/\varepsilon\delta)$ samples per least-square problem, we have*

$$\left\| A^{\neq j}(\tilde{A}_j)^\top_{(2)} - X^\top_{(j)} \right\| \leq (1+\varepsilon) \min_{(A_j)_{(2)}} \left\| A^{\neq j}(A_j)^\top_{(2)} - X^\top_{(j)} \right\|,$$

*with probability $(1 - \delta)$, where $\tilde{A}_j$ is the solution of the sketched least-squares problem, for all least-squares solve. The efficient sampling procedure of Theorem 1.1 brings the overall complexity to $\tilde{O}\left( \frac{\#it}{\varepsilon\delta} R^4 \cdot \sum_{j=1}^{N} N \log I_j + I_j \right)$, where "#it" is the number of ALS iterations.*

Algorithms 1 and 2 refer to procedures "BuildSampler" and "RowSample" that were first used to sample from the Khatri-Rao product. The $k$-th BuildSampler data structure creates a full binary tree that truncated to $\log\lceil I_k R_{k-1}/R_k \rceil$ levels, each caching an $R_k \times R_k$ matrix containing information from $A_k^L$. To draw a sample, the RowSample procedure executes a random walk from the root to the leaves requiring $O(R_k^2)$ work at each internal node. The sampler performs matrix-vector multiplication with the cached data at each internal node and compares the output value to a threshold, using the comparison to branch either left or right in the random walk. Assuming $R_1 = ... = R_j$ and $I = I_1 = ... = I_j$, the storage cost of each sampler is $O(IR^2)$ (computed by multiplying the matrix size stored at each node by the maximum node count). By multiplying the tree depth by the matrix-vector multiplication cost at each node, we get runtime cost $O(R^2 \log I)$ to draw one sample from $A_k^L$. For the motivation behind the procedure, details of its correctness, and pseudocode, we refer the reader to the original work [Bharadwaj et al., 2023].

# 5 Experiments

All experiments were conducted on CPU nodes of the NERSC Perlmutter, an HPE Cray EX supercomputer, and the Mila Quebec AI Institute compute cluster. Our code is available at `https://github.com/vbharadwaj-bk/ortho_tt_subspace_embedding`. In this section, we demonstrate the effectiveness of the proposed rTT-ALS on two types of tensors: (i) synthetic and real dense datasets and (ii) real sparse datasets. We use the fit as evaluation metric (higher is better): $\text{fit}(\tilde{\mathcal{X}}, \mathcal{X}) = 1 - \|\tilde{\mathcal{X}} - \mathcal{X}\|_F / \|\mathcal{X}\|_F$, where $\tilde{\mathcal{X}}$ is the TT approximation and $\mathcal{X}$ is the target tensor. The goal of the dense tensor experiments is to show that rTT-ALS has a better time complexity than TT-ALS and TT-SVD while matching rTT-SVD in terms of fit. The sparse tensor experiments show that SVD-based decompositions cannot handle high-order (sparse) tensors. We compare rTT-ALS with the classical TT-ALS. The runtime improvements are most significant for large sparse tensors. Figure 4 compares accuracy (y-axis, higher is better) against ALS iteration time, for rTT-ALS versus non-randomized ALS. The speedup per iteration can be as high as 26x for lower ranks. Particularly, for the NELL-2 tensor, the plot shows that accuracy within three significant figures of non-randomized ALS was achieved roughly 3-4x faster than an optimized non-randomized ALS baseline.

## 5.1 Decomposition of Synthetic and Real Dense Datasets

We compare rTT-ALS to three other methods; TT-SVD [Oseledets, 2011], Randomized TT-SVD (rTT-SVD) [Huber et al., 2017] and TT-ALS [Holtz et al., 2012]. We use TensorLy [Kossaifi et al., 2019] for SVD-based methods and our own implementation for deterministic TT-ALS. For simplicity, we set $R_1 = \cdots = R_{N-1} = R$ for all experiments. For all algorithms, we illustrate the quality of performance by fit and runtime.

**Synthetic Data Experiments.** For the synthetic data experiment, we generate random tensors of size $I \times I \times I$ for $I \in \{100, \ldots, 500\}$ and of TT rank $R = 20$ (by drawing each core's components i.i.d. from a standard normal distribution). A small Gaussian noise with mean zero and standard deviation of $10^{-6}$ is added to each entry of the resulting tensor. We then run the four methods to find a rank $\tilde{R} = 5$ approximation of the target tensor. ALS-based methods are initialized using their SVD-based counterpart (TT-ALS with the output of TT-SVD and rTT-ALS with the output of rTT-SVD) and are run for 15 iterations. The sample count for rTT-ALS is fixed to $J = 5000$ for all values of $I$. The average fit over 5 trials for all four algorithms are reported as a function of the dimension in Figure 3. rTT-ALS is about $2\times$ faster than TT-ALS and $3\times$ faster than TT-SVD for $I = 500$. Although rTT-SVD is the fastest method, it achieves poor performance in terms of fit.

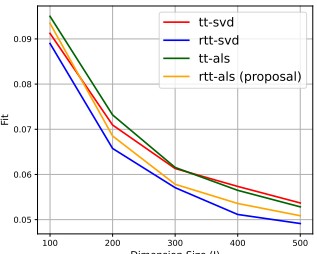 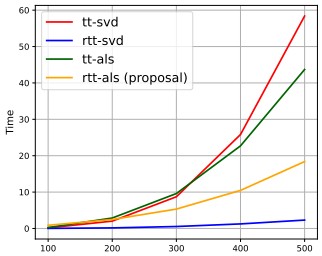

Figure 3: Fit (left) for $J = 5000$ and running time (right) averaged over 5 trials for the synthetic data experiment.

Table 1: Decomposition results for real datasets with $J = 2000$ and the target rank $\tilde{R} = 5$. Time is in seconds.

| Method | Pavia Uni. | | Tabby Cat | | MNIST | | DC Mall | |
| | Fit | Time | Fit | Time | Fit | Time | Fit | Time |
| --- | --- | --- | --- | --- | --- | --- | --- | --- |
| TT-ALS | 0.61 | 4.16 | 0.65 | 44.570 | 0.46 | 8.29 | 0.59 | 21.86 |
| **rTT-ALS (proposal)** | 0.60 | 0.82 | 0.65 | 7.360 | 0.45 | 2.20 | 0.59 | 2.81 |
| TT-SVD | 0.61 | 6.65 | 0.65 | 136.189 | 0.46 | 17.19 | 0.59 | 41.45 |
| rTT-SVD | 0.61 | 0.33 | 0.65 | 4.285 | 0.46 | 0.65 | 0.59 | 0.46 |

**Real Data Experiments.** For the real data experiment, we consider four real images and video datasets (more details about datasets are given in Appendix C): (i) Pavia University is a hyper-spectral image dataset of size $(610 \times 340 \times 103)$, (ii) DC Mall is also a dataset of hyper-spectral images of size $(1280 \times 307 \times 191)$. Both datasets are three-dimensional tensors where the first two dimensions are the image height and width, and the third dimension is the number of spectral bands, (iii) the MNIST dataset is of size $(60000 \times 28 \times 28)$, and iv) Tabby Cat is the three-dimensional tensor of size $(720 \times 1280 \times 286)$ which contains grayscale videos of a man sitting on a park bench and a cat, respectively. The first two dimensions are frame height and width, and the third dimension is the number of frames. For all datasets, the preprocessing step is done by tensorizing data tensors into higher-dimensional tensors. Table 1 illustrates the results for a single trial when $\tilde{R} = 5$. For all datasets we keep the sample count fixed to $J = 2000$. Similarly to the synthetic data experiments, rTT-ALS is faster than TT-ALS and TT-SVD (up to $10\times$ faster than TT-ALS).

## 5.2 Approximate Sparse Tensor Train Decomposition

We next apply rTT-ALS to three large sparse tensors from FROSTT [Smith et al., 2017]. Table 2 gives the fits achieved by our method to decompose these tensors. The largest of these tensors, NELL-2, has around 77 million nonzero entries with mode sizes in the tens of thousands. Fits for sparse tensor decomposition are typically low, but the factors of the resulting decomposition have

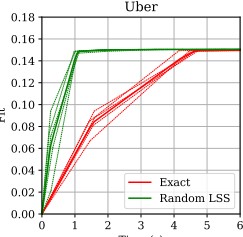 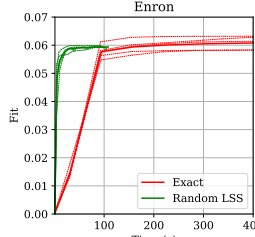 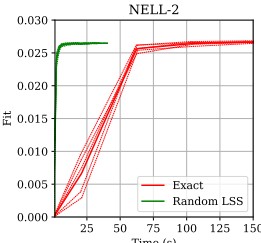

Figure 4: Fit as a function of time for three FROSTT tensors, $R = 6$, $J = 2^{16}$ for rTT-ALS. Thick lines are averages of 5 fit-time traces, shown by thin dotted lines.

Table 2: Average fits and speedup, $J = 2^{16}$ for randomized algorithms, 40 ALS iterations. The speedup is the average per-iteration runtime for a single exact ALS sweep divided by the average time for a single randomized sweep.

| | Uber | | | Enron | | | NELL-2 | | |
|---|---|---|---|---|---|---|---|---|---|
| $R$ | rTT-ALS | TT-ALS | Speedup | rTT-ALS | TT-ALS | Speedup | rTT-ALS | TT-ALS | Speedup |
| 4 | 0.1332 | 0.1334 | 4.0x | 0.0498 | 0.0507 | 17.8x | 0.0213 | 0.0214 | 26.0x |
| 6 | 0.1505 | 0.1510 | 3.5x | 0.0594 | 0.0611 | 12.4x | 0.0265 | 0.0269 | 22.8x |
| 8 | 0.1646 | 0.1654 | 3.0x | 0.0669 | 0.0711 | 10.5x | 0.0311 | 0.0317 | 22.2x |
| 10 | 0.1747 | 0.1760 | 2.4x | 0.0728 | 0.0771 | 8.5x | 0.0350 | 0.0359 | 20.5x |
| 12 | 0.1828 | 0.1846 | 1.5x | 0.0810 | 0.0856 | 7.4x | 0.0382 | 0.0394 | 15.8x |

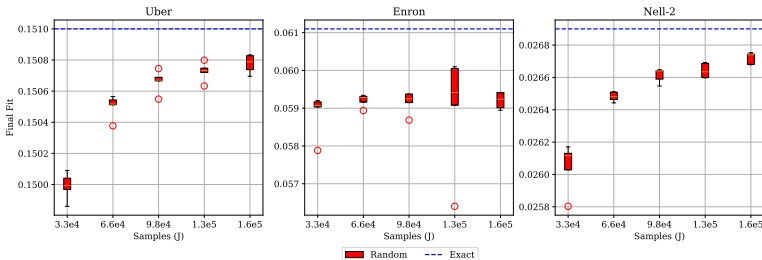

Figure 5: Final fit of sparse tensor decomposition for varying sample counts. Each boxplot reports statistics for 5 trials. The blue dashed lines show the fit for non-randomized ALS.

successfully been mined for patterns [Larsen and Kolda, 2022]. For these experiments, we chose all decomposition ranks equal with $R_1 = ... = R_N = R$ and tested over a range of values for $R$.

The fits produced by rTT-ALS match those produced by the non-randomized ALS method up to variation in the third significant figure for Uber and NELL-2, with slightly higher errors on the Enron tensor. We kept the sample count for our randomized algorithms fixed at $J = 2^{16}$ throughout this experiment. As a result, the gap between the fit of the randomized and exact methods grows as the target rank increases, which our theory predicts.

Table 2 also reports the average speedup per ALS sweep of rTT-ALS over the exact algorithm. On the NELL-2 sparse tensor with target rank 12, the non-randomized ALS algorithm requires an average of 29.4 seconds per ALS sweep, while rTT-ALS requires only 1.87 seconds. Figure 4 shows that our method makes faster progress than its non-randomized counterpart across all three tensors. Because we could not find a well-documented, high-performance library for sparse tensor train decomposition, we wrote a fast multithreaded implementation in C++, which serves as the baseline method in these figures and tables (the code will be made publicly available).

Figure 5 shows the impact of varying the sample count on the final fit. We find modest increases in accuracy for both Uber and NELL-2 as the sample count increases by a factor of 5 (starting from $J = 2^{15}$). Increasing $J$ has a smaller impact for the Enron tensor, which is generally more difficult to decompose beginning with i.i.d. random factor initialization [Larsen and Kolda, 2022].

## 6 Conclusion

We proposed a sampling-based ALS method leveraging an efficient data structure to sample from the exact leverage scores. More precisely, we show that by exploiting the canonical form of the TT decomposition, leverage scores can be computed efficiently for all the least squares problems of ALS. We provide strong theoretical guarantees for the proposed data structure. Experiments on massive dense and sparse tensors confirm the theoretical results. The sampling algorithm we proposed could be extended to more general tree-based tensor network structures, leveraging canonical forms in a similar spirit to rTT-ALS.

## Acknowledgments and Disclaimers

V. Bharadwaj was supported by the U.S. Department of Energy, Office of Science, Office of Advanced Scientific Computing Research, Department of Energy Computational Science Graduate Fellowship under Award Number DE-SC0022158. This research used resources of the National Energy Research Scientific Computing Center, a DOE Office of Science User Facility, under Contract No. DE-AC02-05CH11231 using NERSC award ASCR-ERCAP0024170. BT. Rakhshan was supported by an IVADO Fellowship. O. A. Malik contributed to this work while he was affiliated with Lawrence Berkeley National Laboratory where he was supported by the Office of Science of the DOE under Award Number DE-AC02-05CH11231. G. Rabusseau acknowledges the support of the Canadian Institute for Advanced Research (CIFAR AI chair program) and the Natural Sciences and Engineering Research Council of Canada (Discovery program, RGPIN-2019-05949).

This report was prepared as an account of work sponsored by an agency of the United States Government. Neither the United States Government nor any agency thereof, nor any of their employees, makes any warranty, express or implied, or assumes any legal liability or responsibility for the accuracy, completeness, or usefulness of any information, apparatus, product, or process disclosed, or represents that its use would not infringe privately owned rights. Reference herein to any specific commercial product, process, or service by trade name, trademark, manufacturer, or otherwise does not necessarily constitute or imply its endorsement, recommendation, or favoring by the United States Government or any agency thereof. The views and opinions of authors expressed herein do not necessarily state or reflect those of the United States Government or any agency thereof.

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

# A  Additional Notation

**Definition A.1.** *Let $A \in \mathbb{R}^{m \times n}$ and $B \in \mathbb{R}^{p \times q}$ then the Kronecker product, $A \otimes B \in \mathbb{R}^{mp \times nq}$ is defined by*

$$A \otimes B = \begin{bmatrix} a_{11}B & a_{12}B & \ldots & a_{1n}B \\ a_{21}B & a_{22}B & \ldots & a_{2n}B \\ \vdots & \vdots & \ddots & \vdots \\ a_{m1}B & a_{m2}B & \ldots & a_{mn}B \end{bmatrix}$$

**Definition A.2.** *Let $A \in \mathbb{R}^{m \times R}$ and $B \in \mathbb{R}^{n \times R}$ then the Khatri-Rao product, $A \odot B \in \mathbb{R}^{mn \times R}$ is defined by*

$$A \odot B = \begin{bmatrix} | & | & & | \\ a_1 \otimes b_1 & a_2 \otimes b_2 & \ldots & a_R \otimes b_R \\ | & | & & | \end{bmatrix}$$

*where $a_1, \ldots, a_R \in \mathbb{R}^m$ are the columns of $A$, $b_1, \ldots, b_R \in \mathbb{R}^n$ are the columns of $B$ and the columns of $A \odot B$ is the subset of the Kronecker product. In the corresponding tensor network diagram, the copy tensor captures the fact that the second indices are the same.*

## A.1  Details about Orthogonalization of the TT Decomposition

Figure 6 illustrates the single-site TT-ALS method, which begins with a TT decomposition in canonical form initialized by a crude guess. Core $\mathcal{A}_1$ of the decomposition is non-orthogonal; in sweeps from left-to-right and right-to-left, the algorithm holds all but one core constant and solves for the optimal value for the remaining core. After updating each core, by a QR decomposition the non-orthonormal part is merged to the left or right (depending on the direction of the sweep), a step which is called *core orthogonalization*.

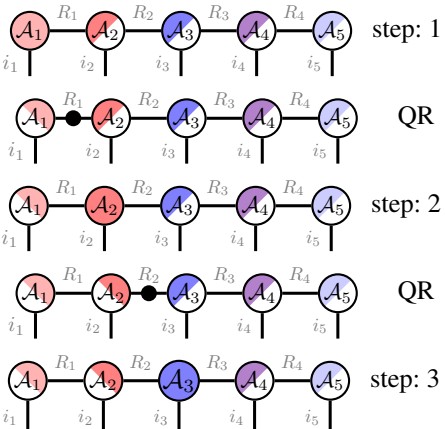

Figure 6: Half-sweep of TT-ALS. In each non-QR step the fully colored core is optimized and in each QR step the non-orthogonal component (depicted by black circle) is absorbed to the next core. This procedure repeats until reaching the right side of the decomposition then the same procedure is repeated from right until reaching to the left side (not demonstrated in this figure.)

# B Proofs

## B.1 Proof of Lemma 4.1

Noting that $A_{\leq j}[\underline{s_1 \dots s_j}, :]$ is a **row vector**, we write

$$
\begin{aligned}
&p(\hat{s}_k = s_k \mid \hat{s}_{>k} = s_{>k}) \\
&= \sum_{s_1, \dots, s_{k-1}} p(\hat{s}_1 = s_1 \wedge \dots \wedge \hat{s}_j = s_j) \\
&= \sum_{s_1, \dots, s_{k-1}} \frac{1}{R_j} \left( A_{\leq j}[\underline{s_1 \dots s_j}, :] \cdot A_{\leq j}[\underline{s_1 \dots s_j}, :]^\top \right) \\
&= \sum_{s_1, \dots, s_{k-1}} \frac{1}{R_j} \mathrm{Tr} \left[ A_{\leq j}[\underline{s_1 \dots s_j}, :]^\top \cdot A_{\leq j}[\underline{s_1 \dots s_j}, :] \right] \\
&= \frac{1}{R_j} \sum_{s_1, \dots, s_{k-1}} \mathrm{Tr} \left[ \mathcal{A}_j[:, s_j, :]^\top \cdot \dots \cdot \mathcal{A}_1[:, s_1, :]^\top \cdot \mathcal{A}_1[:, s_1, :] \cdot \dots \cdot \mathcal{A}_j[:, s_j, :] \right] \\
&= \frac{1}{R_j} \sum_{s_2, \dots, s_{k-1}} \mathrm{Tr} \left[ \mathcal{A}_j[:, s_j, :]^\top \cdot \dots \cdot \left( \sum_{s_1} \mathcal{A}_1[:, s_1, :]^\top \cdot \mathcal{A}_1[:, s_1, :] \right) \cdot \dots \cdot \mathcal{A}_j[:, s_j, :] \right] \\
&= \frac{1}{R_j} \sum_{s_2, \dots, s_{k-1}} \mathrm{Tr} \left[ \mathcal{A}_j[:, s_j, :]^\top \cdot \dots \cdot \mathcal{A}_2[:, s_2, :]^\top \cdot I \cdot \mathcal{A}_2[:, s_2, :] \cdot \dots \cdot \mathcal{A}_j[:, s_j, :] \right].
\end{aligned}
\tag{7}
$$

In the expressions above, the summation over each variable $s_t$, $1 \leq t \leq k$, is taken over the range $[I_t]$. The first step follows by marginalizing over random variables $\hat{s}_1, \dots, \hat{s}_{k-1}$. The second step follows from Equation (4). The third step rewrites an inner product of two vectors as the trace of their outer product. The fourth step follows from the definition of $A_{\leq j}$. The fifth step follows from the linearity of the trace by moving the summation over $s_1$ into the product expression. The last step follows from the definition of the left-orthonormality property on $\mathcal{A}_1$; that is, $\sum_{s_1} \mathcal{A}_1[:, s_1, :]^\top \cdot \mathcal{A}[:, s_1, :] = A_1^{L\top} A_1^L = I$. By successively moving summation operators into the product expression to repeat the last step (exploiting the left-orthonormality of each core in the process), we find

$$
\begin{aligned}
&p(\hat{s}_k = s_k \mid \hat{s}_{>k} = s_{>k}) \\
&= \frac{1}{R_j} \mathrm{Tr} \left[ \mathcal{A}_j[:, s_j, :]^\top \cdot \dots \cdot \mathcal{A}_k[:, s_k, :]^\top \cdot \mathcal{A}_k[:, s_k, :] \cdot \dots \cdot \mathcal{A}_j[:, s_j, :] \right] \\
&= \frac{1}{R_j} \mathrm{Tr} \left[ H_{>k}^\top \cdot \mathcal{A}_k[:, s_k, :]^\top \cdot \mathcal{A}_k[:, s_k, :] \cdot H_{>k} \right],
\end{aligned}
\tag{8}
$$

where the last line follows from the definition of $H_{>k}$. $\qquad \square$

## B.2 Proof of Lemma 4.2

We write

$$
\begin{aligned}
p(\hat{s}_k = s_k \mid \hat{s}_{>k} = s_{>k}) &= \frac{1}{R_j} \mathrm{Tr} \left[ H_{>k}^\top \cdot \mathcal{A}_k[:, s_k, :]^\top \mathcal{A}_k[:, s_k, :] \cdot H_{>k} \right] \\
&= \frac{1}{R_j} \sum_{r=1}^{R_j} \left( e_r^\top \cdot H_{>k}^\top \cdot \mathcal{A}_k[:, s_k, :]^\top \cdot \mathcal{A}_k[:, s_k, :] \cdot H_{>k} \cdot e_r \right) \\
&= \frac{1}{R_j} \sum_{r=1}^{R_j} \left( h_{>k}^\top \cdot \mathcal{A}_k[:, s_k, :]^\top \cdot \mathcal{A}_k[:, s_k, :] \cdot h_{>k} \right) \\
&= \frac{1}{R_j} \sum_{r=1}^{R_j} p(\hat{t}_k = t_k \mid \hat{t}_{>k} = t_{>k}, \hat{r} = r).
\end{aligned}
\tag{9}
$$

The first step follows from Lemma 4.1. The second step follows from the definition of the trace. The third step follows from the definitions of $h_{>k}$ and $H_{>k}$. The fourth step follows from the definition of the variables $\hat{t}_1, \dots, \hat{t}_j$. Now observe that $p(\hat{r} = r) = 1/R_j$ for $1 \leq r \leq R_j$, so we can write

$$p(\hat{s}_k = s_k \mid \hat{s}_{>k} = s_{>k}) = \sum_{r=1}^{R_j} p(\hat{t}_k = t_k \mid \hat{t}_{>k} = t_{>k}, \hat{r} = r)p(\hat{r} = r) \tag{10}$$

$$= p(\hat{t}_k = t_k \mid \hat{t}_{>k} = t_{>k}),$$

which completes the proof. $\square$

## B.3 Efficient Sampling Data Structure

Lemma 4.3 first appeared as Lemma 3.2 in the original work by Bharadwaj et al. [2023]. We state a condensed form of the original claim below:

**Lemma B.1** (Bharadwaj et al. [2023], Original). *Given $U \in \mathbb{R}^{M \times R}$, $Y \in \mathbb{R}^{R \times R}$ with $Y$ p.s.d., there exists a data structure parameterized by positive integer $F$ that requires $O(MR^2)$ time to construct and additional space space $O(R^2 \lceil M/F \rceil)$. After construction, the data structure can draw a sample from the distribution defined elementwise by*

$$q_{h,U,Y}[s] := C^{-1} U[s,:] \left( Y \circledast hh^\top \right) U[s,:]^\top$$

*in time $O(R^2 \log\lceil M/F \rceil + FR^2)$. When $Y$ is a rank-1 matrix, the runtime drops to $O(R^2 \log\lceil M/F \rceil + FR)$.*

In the statement above, $C$ is an appropriate normalization constant. To prove our adapted lemma, take $Y = [1]$, a matrix of all ones that is rank-1, and set $F = R$. Then

$$q_{h,U,Y}[s] = C^{-1} U[s,:] \left( hh^\top \right) U[s,:]^\top = C^{-1}(U[s,:] \cdot h)^2$$

This is the target probability distribution of Lemma 4.3, and the runtime to draw each sample is $O(R^2 \log(M/R) + R^2) = O(R^2 \log(M/R))$. The choice $F = R$ also induces space usage $O(MR)$, linear in the size of the input. Our modified claim follows. $\square$

## B.4 Proof of Theorem 1.1

We provide a short end-to-end proof that shows that Algorithms 1 and 2 correctly draw samples from $A_{\leq j}$ (the matricization of the left-orthogonal core chain) according to the distribution of its squared row norms while meeting the runtime and space guarantees of Theorem 1.1.

**Construction Complexity**: The cost of Algorithm 1 follows from 4.3 with $M = IR_{k-1}$, the row count of $A_k^L$ for $1 \leq k \leq j$. Using this lemma, construction of each sampling data structure $Z_k$ requires time $O(I_k R_{k-1} R_k^2)$. The space required by sampler $Z_k$ is $O(I_k R_{k-1} R_k)$; summing over all indices $k$ gives the construction claim in Theorem 1.1.

**Sampling Complexity:** The complexity to draw samples in Algorithm 2 is dominated by calls to the RowSample procedure, which as discussed in Section 4 is $O(R_k^2 \log(I_k R_{k-1}/R_k))$ Summing the complexity over indices $1 \leq k < j$ yields the cost claimed by Theorem 1.1 to draw a single sample. The complexity of calling the RowSample procedure repeatedly dominates the complexity to update the history vector $h$ over all loop iterations, which is $O\left(\sum_{k=1}^{j} R_{k-1} R_k\right)$ for each sample.

**Correctness:** Our task is to show that Algorithm 2 each sample $t_d$, $1 \leq d \leq J$, is a multi-index that follows the squared row norm distribution on the rows of $A_{\leq j}$. To do this, we rely on lemmas proven earlier. For each sample, the variable $\hat{r}$ is a uniform random draw from $[R_j]$, and $h$ is initialized to the corresponding basis vector. By Equation (6) and Lemma 4.3, Line 5 from Algorithm 2 draws each index $\hat{t}_k$ correctly according to the probability distribution specified by Equation (5). The history vector is updated by Line 6 of the algorithm so that subsequent draws past iteration $k$ of the loop are also drawn correctly according to Equation (5). Lemma 4.2 (relying on Lemma 4.1) shows that the multi-index $\hat{t}_1 \ldots \hat{t}_j$ drawn according to Equation (5) follows the same distribution as $\hat{s}_1 \ldots \hat{s}_j$, which was defined to follow the squared norm distribution on the rows of $A_{\leq j}$. This completes the proof. $\square$

## B.5 Proof of Corollary 4.4

Since $A^{\neq j} \in \mathbb{R}^{\prod_{k \neq j}^{N} I_k \times R_{j-1} R_j}$ and $X_{(j)} \in \mathbb{R}^{\prod_{k \neq j}^{N} I_k \times I_j}$, we draw $\tilde{O}(R^2/\varepsilon\delta)$ samples to achieve the error bound $(1 + \varepsilon)$ with probability $(1 - \delta)$ for each least squares solve in the down-sampled problem (3.5). By Theorem 1.1, the complexity of drawing $J$ samples with our data structure is

$$O\left(\sum_{k \neq j} J \log I_k R^2\right) = \tilde{O}\left(\sum_{k \neq j} R^4/(\varepsilon\delta) \log I_k\right)$$

where we suppose that $R_1 = R_2 = \cdots = R_{N-1}$ and $I_1 = \cdots = I_N$. The cost of sampling a corresponding subset of $X_{(j)}$ is $O(JI_j) = \tilde{O}\left(R^2/(\varepsilon\delta)I_j\right)$. Solving the downsampled least squares problem also costs $O(JR^2 I_j) = \tilde{O}\left(I_j R^4/(\varepsilon\delta)\right)$. Summing them all together for $1 \leq j \leq N$ gives

$$\tilde{O}\left(1/\varepsilon\delta\left(\sum_{j=1}^{N}\left(\sum_{k \neq j} R^4 \log I_k\right) + R^4 I_j\right)\right)$$

$$= \tilde{O}\left(R^4/\varepsilon\delta \cdot \sum_{j=1}^{N}(N-1)\log I_j + I_j\right)$$

$$= \tilde{O}\left(R^4/\varepsilon\delta \cdot \sum_{j=1}^{N} N \log I_j + I_j\right)$$

where we wrote the last equation considering the fact that $N$ dominates $(N - 1)$.

## C Details about Datasets & Experiments

### C.1 Datasets

For the real dense datasets experiment, we truncated and reshaped the original data tensors in to the fourth order tensors as follows.

- **Pavia University dataset**: The original has dimensions $(610, 340, 103)$. We truncate it to $(600, 320, 100)$, permute the modes to dimensions $(100, 320, 600)$ tensor and reshape it into a tensor of dimensions $(100, 320, 24, 25)$. It is available at
  http://lesun.weebly.com/hyperspectral-data-set.html
- **Tabby Cat dataset** is permuted to $(286, 720, 1280)$ and reshaped to a tensor of size $(286, 720, 40, 32)$. The video is in color and converted to grayscale by averaging the three color channels. It is available at
  https://www.pexels.com/video/video-of-a-tabby-cat-854982/.
- **The MNIST dataset** was reshaped into a tensor of size $(280, 600, 28, 10)$ and is available at https://www.kaggle.com/datasets/hojjatk/mnist-dataset
- **The Washington DC Mall dataset** was truncated to dimensions $(1280, 306, 190)$ before reshaping into a tensor of size $(1280, 306, 10, 19)$. It is available at
  https://engineering.purdue.edu/Ëœbiehl/MultiSpec/hyperspectral.html.

The sparse tensors **Uber**, **Enron**, and **NELL-2** were downloaded from the FROSTT collection [Smith et al., 2017]. The dimensions of these tensors were unchanged from the versions available online. Consistent with established practice [Larsen and Kolda, 2022], we computed the logarithm of the tensor values in the Enron and NELL-2 datasets before performing our experiments.

### C.2 Computing Resources

The dense data experiments were conducted on MILA cluster nodes with 4 CPUs and 16GB of RAM each. Sparse tensor decomposition experiments were conducted on NERSC Perlmutter nodes with 2 CPUs and 512 GB of RAM each.

