# OpenReview forum: "Efficient Leverage Score Sampling for Tensor Train Decomposition"
_NeurIPS.cc/2024/Conference — NeurIPS 2024 poster_

### Official Review · Reviewer_8ErW · 2024-07-07

**Soundness:** 3
**Presentation:** 4
**Contribution:** 3
**Rating:** 7
**Confidence:** 3

**Summary:**

This paper gives a better randomized alternating least squares algorithm for computing tensor factorizations. It's based on an exact characterization of leverage scores of the matrization of tensors via a suitable intermediate orthonormal representation. This is justified rigorously, and significant empirical gains were demonstrated.

**Strengths:**

The approach taken is natural, and the bounds obtained are quite powerful.

The experiments considered both synthetic and real data sets, and demonstrated clear gains in the parameter regimes considered.

**Weaknesses:**

I'm a bit concerned about the setting of the experiments, which seem to be 3-dimensional dense tensors. My understanding is that a lot of the more complicated tensor instances are sparse and in higher dimensions. However, I'm not sure whether those have low rank representations.

**Questions:**

Would it be possible to check how the orthogonality conditions are maintained in intermediate steps under the inexact arithmetic caused by round off errors? Aka. are the conditions for the characterizations of leverage scores preserved exactly? (this has been answered)

**Limitations:**

yes

---

> ### Author Rebuttal · Authors · 2024-08-06
>
> We thank the reviewer for their comments and feedback, please find responses below:
>
> **Weaknesses:**
> > I'm a bit concerned about the setting of the experiments, which seem to be 3-dimensional dense tensors. My understanding is that a lot of the more complicated tensor instances are sparse and in higher dimensions. However, I'm not sure whether those have low rank representations.
>
> Correct. We added dense synthetic and real data experiments to show that our proposed approach has a better time complexity than TT-ALS and TT-SVD and works slightly better than rTT-SVD in terms of fit. The main point is to demonstrate that our new approach works as well as other types of TT decompositions for dense data. As SVD-based methods cannot handle high-dimensional sparse tensors. For the more complicated tensors, we added a sparse tensor data experiment as you can see in the second part, the setting is different and more complicated as SVD-based TT decomposition cannot be used and we just compared our approach with the classical TT-ALS method.
>
> **Questions:**
> > Would it be possible to check how the orthogonality conditions are maintained in intermediate steps under the inexact arithmetics caused by round off errors? Aka. are the conditions for the characterizations of leverage scores preserved exactly?
>
> We did not observe any numerical instabilities due to a round of errors during the orthogonalization step (which is performed using the stable QR decomposition routine provided by numpy.linalg) and we believe the characterization of leverage scores is preserved exactly (up to machine precision).

---

> > ### Comment · Reviewer_8ErW · 2024-08-12
> > **thank you**
> >
> > Thank you for the detailed responses.
> >
> > My concern w.r.t. roundoffs is that equality conditions like equation (6) might not be robust to perturbations to the original matrix. However, I now see that it's only for the initial matrix, not for the intermediate products of sampling. So I agree that it's inherently robust.
> >
> > I will raise my presentation score, and leave the overall unchanged.

---

### Official Review · Reviewer_oeCx · 2024-07-11

**Soundness:** 3
**Presentation:** 3
**Contribution:** 2
**Rating:** 6
**Confidence:** 4

**Summary:**

The authors proposed a leverage score sampling-based TT-ALS method to reduce the computational complexity of the traditional TT-ALS. Experimental results verify the performance of the proposed method.

**Strengths:**

The paper is well written with good theoretical analysis and desired experimental performance. The method using leveraging score sampling is technically sound for TT decomposition.

**Weaknesses:**

The contribution and novelty are not clearly stated compared with [Malik and Becker, 2021]. In [Malik and Becker, 2021], the leverage sampling was applied to TR. As TT can be treated as a special case of TR, the author does not clearly state that it is necessary to develop a new method for TT, or that there are some new theories or findings that are distinct from the previous TR structure.

**Questions:**

1. My main concern about this work is its relation to the leverage score sampling-based TR-ALS [Malik and Becker, 2021]. As TT can be seen as a special case of TR, what is the main contribution of the current work compared with leverage score sampling-based TR-ALS? Please clarify that it is necessary to develop a new method beyond [Malik and Becker, 2021] (i.e. has a better theoretical guarantee or has better performance than leverage score sampling-based TR-ALS).

2. For real data experiments, how about the performance compared with other tensor decomposition structures, such as TR? As you can set the first and last TR-rank as 1 thus it reduces to a TT-like structure.

---

> ### Author Rebuttal · Authors · 2024-08-06
>
> We thank the reviewer for their comments and feedback, please find responses below:
>
> **Weaknesses:**
> > The contribution and novelty are not clearly stated compared with [Malik and Becker, 2021]. In [Malik and Becker, 2021], the leverage sampling was applied to TR. As TT can be treated as a special case of TR, the author does not clearly state that it is necessary to develop a new method for TT, or that there are some new theories or findings that are distinct from the previous TR structure.
>
> In [Malik et al, 2021], the leverage scores were approximated using a product of simpler distributions. They sample according to the leverage scores of each core. While in our paper, we proposed a novel data structure that can compute the exact leverage scores. Moreover, the time complexity to solve one least-square of ALS with the method of Malik et al.  Malik et al approach is $NIR^4$ +#iter. $NIR^{(2N+2)}/(\epsilon\delta)$, which is still exponential in the order of the tensor $N$, while our result does not have any exponential dependency to the order of the tensor. We will add and emphasize in the camera-ready version that our approach differs from theirs in two main aspects:
> - We did not use approximation for finding the leverage scores, instead a novel data structure is proposed for finding **exact** leverage scores.
> - In Malik et al, the runtime for the least-square solve still has an exponential dependency to the order of a tensor while our approach is free of this exponential dependency.
>
> **Questions**
> > My main concern about this work is its relation to the leverage score sampling-based TR-ALS [Malik and Becker, 2021]. As TT can be seen as a special case of TR, what is the main contribution of the current work compared with leverage score sampling-based TR-ALS? Please clarify that it is necessary to develop a new method beyond [Malik and Becker, 2021] (i.e. has a better theoretical guarantee or has better performance than leverage score sampling-based TR-ALS).
>
> Indeed, TT is a special case of the TR decomposition. However, in [Malik and Becker, 2021], leverage score approximation is used for the TR-ALS with time complexities #iter · $NIR^{2N+2}/(\epsilon\delta)$ (further improved to and #iter · $N^3R^8 (R + I/\epsilon)/\delta $ in [Malik, 2022]), respectively. In contrast, our method has a lower time complexity and provides a better theoretical guarantee (please see corollary 4.4 in our paper). Therefore, our method goes beyond the works of Malik et al. As mentioned in their works, computing leverage scores requires computing the left singular values which has the same cost as solving the original least squares problem. For this reason, they approximate the leverage scores. Instead, we propose a novel data structure to sample from the **exact** leverage scores at a lower cost than their proposed approaches.
>
> > For real data experiments, how about the performance compared with other tensor decomposition structures, such as TR? As you can set the first and last TR-rank as 1 thus it reduces to a TT-like structure.
>
> It is true that TT is a special case of TR, but note that the main problem we address in our paper is how to scale TT-ALS to very large tensors using randomized techniques. The purpose of our experiments is to demonstrate how we efficiently achieve this goal with exact leverage score sampling. Extending our approach to TR-ALS could be interesting, however it would be very challenging due to the lack of canonical forms for TR decomposition.
>
> We hope we have addressed all your concerns and answered your questions  (in which case we kindly ask you to consider increasing your score). We are happy to clarify any additional point during the discussion period.

---

> > ### Comment · Reviewer_oeCx · 2024-08-12
> >
> > Thank the reviewer for their reply. I've increased the score.

---

### Official Review · Reviewer_BTzr · 2024-07-12

**Soundness:** 3
**Presentation:** 2
**Contribution:** 3
**Rating:** 4
**Confidence:** 3

**Summary:**

This paper presents an efficient algorithm to use leverage-score sampling to solve least squares problems arising as subproblems in a larger alternating least squares (ALS) algorithm for building an approximate Tensor Train (TT) decomposition. The paper reports empirical evaluation of the proposed algorithm, showing runtime improvement compare to the baseline un-sampled problems.

**Strengths:**

- Empirical evaluation show a runtime improvement with respect to unsketched TT-ALS, without reducing fit.
- The paper presents an efficient algorithm for leverage score sampling of ALS subproblems in TT-ALS.

**Weaknesses:**

Major comments (affecting recommendation):
- The proposed method is compared empirically only to a small number of baselines. Seems that the most likely competatitor (Chen et al. 2023) is not compared against.
- Runtime improvement in the experiments is modest. This is a sampling based method, so if sampling is very efficient (due to efficient leverage score approximation) I would expect a big improvement in runtime. This might not be the case if either more iterations are now needed for convergence or computing the sampling probabilities is very expensive. Unclear from then experiments which is the case.
- The paper does not present an analysis end-to-end. It only analyzes how to subsample the ALS subproblems, and thus accelerate them. Not analyzed is how the solution of the subsampled system in lieu of the full subproblem affect ALS convergence.
- Many details required for implementation of the algorithm are omitted. Many points regarding the algorithm are not clear.

Minor comments (do not affect the recommendation):
- Lines 125 - 126: add citation.
- Figure 3 left: is this fit in the y-axis or misfit?
- Line 108-109: I think you mean *omission* of multiplicative terms.
- Lines 58-60: shouldn't runtime be O(jIR^3)?
- Line 54: The term "left matricization" is used without defining it yet.
- Line 55: I think it should be R_j
- Eq (2): I think on the right it should be A[i,:]^T. Similar issue with Eq (3).

**Questions:**

- Lines 96-97: Sketching is a good way to circumvent the need to compute leverage scores, and TensorSketch allows us to sketch matrices with Kronecker structure. And indeed, the authors point out that [Chen et al. 2023] do this. How do their results compare to the present paper? Why is what you are proposing a better idea than using TensorSketch?
- Lines 64-66: What is the significance of this observation regarding the complexity of the algorithm?
- Lines 125 - 126: One the canonical form has been computed, can we solve the ALS subproblem efficiently?
- Did you compare empirically your method to the method of [Chen et al. 2023].
- Is your algorithm computing the canonical form anew in each ALS iteration, or are you somehow updating it?
- Does your algorithm maintain/computes the leverage scores of each row (the number of such scores is exponential), or do you present only an efficient way to sample from it without forming it?

**Limitations:**

Nothing to add.

---

> ### Author Rebuttal · Authors · 2024-08-06
>
> We thank the reviewer for their comments and respond to the feedback below:
>
> **Weaknesses:**
> > [W1][Comparison against (Chen et al. 2023)]
>
> The TensorSketch approach by Chen et. al. requires an **exponential** sample count in the tensor dimension q to achieve the $(\epsilon, \delta)$-guarantee on the accuracy of each least-squares solve. See Theorem 4.4 of their work and note the 3^q term in the sample complexity, where (in their notation) q is the tensor dimension. By contrast, our algorithm requires a sample count that has **no dependence** on the tensor dimension q; the sample count depends only on the column count of the design matrix and the parameters $\epsilon$nd $\delta$. The time to draw these samples in our method is linearly proportional to q, the tensoring dimension, a modest cost. The efficiency of our method stems from the fact that we sample from the exact distribution of squared row norms of the design matrix. By contrast, TensorSketch applied without tree structure modifications, such as those proposed by Ahle et al., suffers from an exponential output sketch size. We will clarify this point at the end of Section 2 and in the Introduction.
>
> > [W2][Runtime improvement]
>
> The runtime improvements are most significant for large sparse tensors. Figure 4 shows the accuracy (y-axis, higher is better) vs. ALS iteration time for our method vs. non-randomized ALS. The speedup per iteration can be as high as 26x for lower ranks. For the NELL-2 tensor, the plot shows that accuracy within three significant figures of non-randomized ALS was achieved roughly 3-4x faster than an optimized non-randomized ALS baseline. We will note this at the beginning of Section 5 (Experiments) and will highlight these facts further in Section 5.2.
>
> > [W3][End-to-end analysis]
>
> Correct. Our analysis provides guarantees on the accuracy of individual least-squares problems, in line by prior works by Cheng et. al, Larsen and Kolda, and Malik. Under additional assumptions on the tensor, convergence guarantees can be derived for sketched ALS: see, for example, https://proceedings.mlr.press/v119/gittens20a/gittens20a.pdf. The global guarantees derived in this paper by Aggour, Gittens, and Yener depend on sketching guarantees established for each linear least-squares problem, which we provide in our work.
>
> > [W4][Implementation details]
>
> The revised version of our draft will include (in section 4.1) detailed descriptions of the BuildSampler and RowSample functions, the data structure described in the appendix to draw samples from each orthonormal core flattening. Our Git repository, which is public at https://anonymous.4open.science/r/fast_tensor_leverage-EB01, includes a simple, slow reference implementation written in Python for this data structure, providing sufficient detail to understand and fully replicate our method.
>
> > Minor comments
>
> Thanks for catching these typos, we will correct and address them all. For Figure 3, the y-axis is the fit.
>
> **Questions:**
> >[Q1] [Our approach vs TensorSketch]
>
> We believe we addressed this point in the comment above on the exponential complexity of TensorSketch without appropriate modifications. Our method offers sub-exponential sample complexity and worst-case runtime complexity to achieve guarantees on the solution to each least-squares problem. The approach by Chen. et. al. requires a worst-case sketch size exponential in the tensor dimension.
>
> > [Q2][Complexity of the algorithm]
>
> The significance here is that data structure construction (as well as subsequent updates to the data structure) does not increase the asymptotic complexity of ALS tensor train decomposition.
>
> > [Q3][proceeding only with the canonical form]
>
> Even after computing the canonical form (which makes the design matrix orthonormal for each linear least squares problem orthonormal), we must multiply the matricized tensor against the chain of TT cores placed in canonical form. **This is the major computational bottleneck** - without sketching, the runtime cost for this matrix multiplication scales as  $O(nnz(T) N R^2)$, where $nnz(T)$ is the number of nonzeros in the tensor, $N$ is the order of the tensor, and $R$ is the tensor train rank. The tensor may have hundreds of millions of nonzero values. Sketching allows us to select only a subset of rows from the design matrix and the corresponding subset of rows from the matricized tensor, reducing the cost to $O(N R^2)$ * (number of selected nonzeros extracted by the sketch). We can update our draft to clarify this point in Section 1.1.
>
> > [Q4][Emprical comparison with Chen et al. 2023]
>
> We did not compare empirically with Chen et. al results. To the best of our knowledge, there is no public code available to replicate their results and compare with them. Also, as explained previously, from the theoretical point of view, the method proposed by Chen et. al cannot scale to the size of the tensors that we consider in the large sparse tensors experiments.
>
> > [Q5][Updating canonical form in each iteration]
>
> We update the canonical form without recomputing it entirely at each substep of ALS. After solving for the optimal value for each core, we compute the QR decomposition of its left or right flattening and replace the newly-computed core with the appropriately-reshaped Q factor. In this way, only one core needs to be updated after each least-squares solve to maintain the canonical form.
>
> > [Q6][Computing the leverage scores]
>
> We do not compute the leverage scores of all rows, since, as you pointed out, there are exponentially-many rows in the tensor dimension. To design a computationally-efficient sampling algorithm, we build an efficient data structure to **sample** from this distribution without materializing it.
>
> We hope we have addressed all your concerns and answered your questions (in which case we kindly ask you to consider increasing your score). We are happy to clarify any additional point during the discussion period.

---

> > ### Author Response · Authors · 2024-08-12
> >
> > Dear reviewer, considering that the end of the discussion period is approaching, do you have any further questions? We think we have answered all your concerns in our rebuttal and hope you'll consider increasing your score.

---

> > ### Comment · Reviewer_BTzr · 2024-08-13
> > **[W1][Comparison against (Chen et al. 2023)]**
> >
> > Thank you for the detailed answers and clarifications. It will be beneficial to include many of the clarifications in revised versions of the manuscripts. I will consider how your answers affect my final score, which is "borderline" to begin with.
> >
> > However, let me followup the discussion on Chen et al. 2023. I see your point. However, note that Thm 4.4 in Chen et al. 2023 gives exact constants (no O() expression)  and this is where the 3^q comes from. In contrast, your results is J=O(R^2/(\eps * \delta)) where the constant is unclear. For all I know, the O() hides a 3^q constant! So: if you flesh out the constant, do you really do not have any dependence on the tensor order?
> >
> > Re empirical comparison: even if we accept that there is a theoretical advantage in the form of a dependence of the order, this does not necessarily tell the whole story for a small q (where typically q=3 is the most common use-case). In fact, if q is fixed to 3, then 3^q is now a constant, and it really matters how it compares the constant of the your algorithm.
> >
> > I still hold that Chen et al. 2023 is the most appropriate empirical baseline,  and without it the empirical evaluation is lacking.

---

> > > ### Author Response · Authors · 2024-08-13
> > >
> > > >Thank you for the detailed answers and clarifications. It will be beneficial to include many of the clarifications in revised versions of the manuscripts.
> > >
> > > Thank you again for you review and questions. We will include all the points mentioned in our rebuttal in the revisions which will improve the clarity.
> > >
> > > > However, let me followup the discussion on Chen et al. 2023. I see your point. However, note that Thm 4.4 in Chen et al. 2023 gives exact constants (no O() expression) and this is where the 3^q comes from. In contrast, your results is J=O(R^2/(\eps * \delta)) where the constant is unclear. For all I know, the O() hides a 3^q constant! So: if you flesh out the constant, do you really do not have any dependence on the tensor order?
> > >
> > > There is no hidden dependency on the tensor order in the big O.
> > >
> > > In Corollary 4.4, the number of samples required for each least square problem, J=O(R^2/(\eps * \delta)), is independent on the tensor order N (N is not hidden in the constant). Thus, for one sweep of the ALS algorithm, the complexity is linear in the tensor order, as stated in the second part of the corollary (i.e. the overall ALS complexity is O(#it εδ R^4  N \sum_{j=1^N} (log I_j + I_j), where N is the order of the tensor). We will clarify this point in the revision.
> > >
> > > > even if we accept that there is a theoretical advantage in the form of a dependence of the order, this does not necessarily tell the whole story for a small q (where typically q=3 is the most common use-case). In fact, if q is fixed to 3, then 3^q is now a constant, and it really matters how it compares the constant of the your algorithm.
> > >
> > > We agree that if q is treated as a constant, then the asymptotic complexity is the same. However, we believe that designing an algorithm with only a linear dependency on the order, instead of the exponential one given in Chen et al, constitute a non-trivial contribution and advancement compared to Chen et al. In addition, the TT decomposition is particularly suited and relevant for (very) high order tensors where the tensor order cannot be treated as constant  (as demonstrated by its uses in quantum physics for simulating many-body systems where the order corresponds to the number of particles in the system).
> > >
> > > > I still hold that Chen et al. 2023 is the most appropriate empirical baseline, and without it the empirical evaluation is lacking.
> > >
> > > Unfortunately, the lack of a publicly available code for the method presented in (Chen et al, 2023) did not allow us to include it in the synthetic experiment. Moreover, to the best of our understanding, the method proposed by Chen et. al could not scale to the size of the tensors that we consider in the large sparse tensors experiments.

---

### Official Review · Reviewer_yjyo · 2024-07-12

**Soundness:** 3
**Presentation:** 3
**Contribution:** 3
**Rating:** 7
**Confidence:** 4

**Summary:**

This paper considers the problem computing the tensor train TT decomposition of large tensors and proposes a novel randomized approach for efficiently solving the problem. In particular, the Alternating Least Squares (TT-ALS) algorithm is considered and exact leverage scores sampling approach is proposed to accelerate the approach. A data structure based approach is devised to efficiently compute the leverage scores of the appropriate matricization of the tensors and sample rows from them. Theoretical analysis show the sampling complexity. Numerical experiment results are presented which to illustrate the performance of the proposed algorithm.

**Strengths:**

The strengths of the paper  are:
1. A novel randomized approach is presented for computing the TT decomposition of large tensors.
2. New data structure and sampling approach are proposed. Theoretical results are presented.
3. Numerical results show that the proposed method is efficient.

**Weaknesses:**

The weaknesses of the paper are:
1. The randomized SVD approach seems to be more efficient in terms cost compared the the prosed method.
2. Certain aspects of the paper can be improved (see below).

**Questions:**

The paper presents an interesting approach to compute the TT decomposition of large tensors.
However, I have the following comments that can further improve the paper:

1. r-TT-SVD seems better -

Based on numerical results presented, it appears  the randomized SVD based approach  (r-TT-SVD) takes significantly lower time, yet achieves comparable performance (fit) as other methods. Are TT-ALS and rTT-ALS popular and relevant for TT decomposition. How do these methods differ in the quality of the decompositions computed?

In some cases, rTT-SVD might have a slightly lower fit. But, since the runtime is so low, perhaps a larger sketch size  would yield similar fit at lower cost. Details about the sketch sizes used are missing.


2. Presentation:
Certain aspects of the presentation can be improved, and there are few minor typos.

i. Are TT decompositions popular in tensor applications? Overall, they seem to have higher computational cost and are not very interpretable. Few alternate tensor decompositions achieve better compression with lower cost. A discussion on the motivation for the use of TT decomposition will benefit the paper (readers can better appreciate the results).

ii. Introduction has tensor jargons which might not be known to general AI audience, such as 3D core chains, left-matricization, contraction of tensor, etc. It is better to avoid these or define them before use.

iii. The computational  cost of solving eq (1) can be added.

iv. Theorem 1.1, point 1 has a $j$ missing in the runtime cost.

v. Not sure what is the subscript (2) in Corollary 4.4.

Many of the results in the paper is based on another Arxiv paper (which is not peer-reviewed). So, correctness of these results is not established.

**Limitations:**

See above comments


------
Post Rebuttal:

I have read other reviews and authors' responses to all reviews. The responses have adequately addressed my concerns. The work has many merits. I have raised my score.

---

> ### Author Rebuttal · Authors · 2024-08-06
>
> We thank the reviewer for their comments and feedback, please find responses below:
>
> **Weaknesses:**
> > The randomized SVD approach seems to be more efficient in terms cost compared the the prosed method.
>
> Randomized SVD cannot scale to large tensors. Indeed, to decompose a tensor using randomized SVD at each step we need to produce a Gaussian tensor of size $s_j\times I^{(N-1)}$, where N is the order of the tensor, $I$ is the physical dimension of each mode and $s_j$ is the size of the sketch. This approach works well when $N$ is small. But randomized svd cannot handle very high-order tensors. In the experiment part, we only use tt-svd and randomized tt-svd for the dense tensors with small $N$, but these two algorithms cannot scale to the case of large sparse tensors in the second part of the experiments.
>
> **Questions:**
> > The paper presents an interesting approach to compute the TT decomposition of large tensors. However, I have the following comments that can further improve the paper:
> r-TT-SVD seems better. Based on numerical results presented, it appears the randomized SVD based approach (r-TT-SVD) takes significantly lower time, yet achieves comparable performance (fit) as other methods. Are TT-ALS and rTT-ALS popular and relevant for TT decomposition.
>
> yes : TT-ALS is a very popular algorithm, notably in the quantum physics community, where it is closely related to the density renormalization group (DMRG) algorithm.
> > How do these methods differ in the quality of the decompositions computed?
>
> Regarding the quality of the decomposition, TT-ALS usually finds better solutions than TT-SVD (TT-SVD can actually be used as an initialization for TT-ALS). Note however that the focus of our paper is on improving the TT-ALS algorithm using randomization. Indeed, TT-ALS is a popular algorithm, but, as mentioned previously, the cost of solving the least-squares problems in ALS is exponential in the order of the tensor which is not efficient for decomposing very large tensors. The main goal of our paper is to make the ALS efficient with randomization and doable for high-order tensors. Therefore, the improvement of a TT-ALS algorithm is the main focus of our paper.
>
> > In some cases, rTT-SVD might have a slightly lower fit. But, since the runtime is so low, perhaps a larger sketch size would yield similar fit at lower cost.
>
> That’s correct, by increasing the sketch size we can have a better quality of sketch for rTT-SVD. However, as mentioned earlier, the SVD-based approach is not suitable for decomposing high-order tensors. Even if by increasing the sketch size rTT-SVD still suffers from the curse of dimensionality when N is very large. Therefore, the quality of the sketch of rTT-SVD cannot be compared to rTT-ALS in the case of sparse high-order tensors (section 5.2 second part of the experiments.)
>
> > Details about the sketch sizes used are missing.
>
> In Section 5.1, the sketch sizes are set to $J=5000$ and $J=2000$ for the synthetic and real data, respectively (we mention it in the text but we will add this information in the caption of Figure 3 and Table 1 as well).
>
> > Presentation: Certain aspects of the presentation can be improved, and there are few minor typos.
> i. Are TT decompositions popular in tensor applications? Overall, they seem to have higher computational cost and are not very interpretable. Few alternate tensor decompositions achieve better compression with lower cost. A discussion on the motivation for the use of TT decomposition will benefit the paper (readers can better appreciate the results).
>
> TT decomposition enjoys stable and numerical stabilities while finding rank-r CP decomposition is NP-hard and the number of  parameters of a Tucker decomposition grows exponentially with the order of the tensor. In contrast, the number of parameters is linear in the order of the tensor for both TT and TR decompositions, but TR decomposition is known to suffer numerical stability issues.  For these reasons, TT is more popular in tensors and quantum physics communities (where it is known as Matrix Product States / MPS). To address your concern about motivating the TT format, we will include a more comprehensive motivation in the introduction.
>
> >  ii. Introduction has tensor jargons which might not be known to general AI audience, such as 3D core chains, left-matricization, contraction of tensor, etc. It is better to avoid these or define them before use.
>
> If we understand correctly, your concern is mostly about the main theorem in the introduction which may cause difficulties for other audiences. We will clarify this by clearly stating before the theorem that all relevant definitions are given in Section 3.1.
>
> > iii. The computational cost of solving eq (1) can be added.
>
> Solving Eq (1) exactly (without randomization) has a computational cost of $O(I^N)$. It is mentioned in the introduction but we will emphasize it in the revision in section 3.2.
>
> > iv. Theorem 1.1, point 1 has a 𝑗 missing in the runtime cost.
>
> Correct. We will add j in the final version. Thank you for catching this typo.
>
> > v. Not sure what is the subscript (2) in Corollary 4.4.
>
> This denotes the second mode unfolding of the tensor which is defined in Definition 3.1 in the paper.
>
> > Many of the results in the paper is based on another Arxiv paper (which is not peer-reviewed). So, correctness of these results is not established.
>
> The paper “Fast exact leverage score sampling from Khatri-rao products with applications to tensor decomposition.” by Bharadwaj et el. has been published in Neurips 2023, we will correct the citation in the camera-ready version.

---

### Author Rebuttal · Authors · 2024-08-06

We thank all the reviewers for their comments and feedback, which we all answered in the individual rebuttals below.
We summarize some of our answers to the main points raised by the reviewers (see individual rebuttals for details).

- [SVD-based vs ALS-based approach]: Overall TT-ALS is very popular in the quantum physics community and usually finds better solutions than TT-SVD. Randomized TT-SVD cannot scale to high-order tensors. Each step of the randomized TT-SVD decomposition requires generating a random Gaussian matrix (classical random projection) that only can handle small-order tensors.

- [Comparison with Chen et al, 2023 paper]: The TensorSketch approach proposed by Chen et al, 2023~(https://arxiv.org/pdf/2309.08093) requires **exponential** sketch size in the tensor dimension while our algorithm requires the sketch size that has no dependence on the tensor dimension; the sketch size depends only on the column count of the design matrix and the parameters $\epsilon$ and $\delta$.

- [Comparison with Malik and Becker, 2021 paper]: The approach proposed by Malik et al, 2021~(https://proceedings.mlr.press/v139/malik21b/malik21b.pdf) requires **approximating** the leverage scores. However, in our paper, we proposed a novel data structure that computes the **exact** leverage scores. Moreover, in Malik et al, the runtime for the least-square solve has an **exponential dependency** to the order of a tensor while our approach is **free** of exponential dependency to the order of a tensor.

- [Dense and sparse tensors experiments]: For the experiment part, we added two separate sections, one for the dense and one for the sparse tensors. Our purpose for the dense tensor section is to show that our approach has a better time complexity than TT-ALS and TT-SVD and matches rTT-SVD in terms of fit. However, SVD-based decompositions cannot handle high-order (sparse) tensors.
For the sparse tensors section, we compared our proposed approach with the classical TT-ALS.

We hope we have addressed all your concerns and answered your questions and we are happy to clarify any additional points during the discussion period.

---

### Comment · Area_Chair_DqxZ · 2024-08-12

Reviewers: The author-reviewer discussions end tomorrow (August 13), so please read the authors' rebuttal carefully and respond if you haven't.

---

### Decision · Program_Chairs · 2024-09-25

**Decision:**

Accept (poster)

**Comment:**

After the rebuttal period and inter-reviewer discussion, the final ratings for this paper are {7, 5, 6, 7} since Reviewer BTzr raised their score from 4 to 5, allowing us to reach a unanimous accept decision.

I read the paper myself and am also fine with it being accepted. It's not as crisp as it should be (theory and experiments), but a paper on leverage score-based sampling for Tensor Train (TT) decompositions is missing from "faster tensor decomposition via leverage scores" line of work (CP decomposition in NeurIPS 2016, Tucker decomposition in NeurIPS 2022).